# Turbulence with Magnetic Helicity That Is Absent on Average

Axel Brandenburg [1,2,3,4,*] and Gustav Larsson [1,2]

1. Nordita, KTH Royal Institute of Technology and Stockholm University, Hannes Alfvéns väg 12, 10691 Stockholm, Sweden; gula1839@student.su.se
2. Oskar Klein Centre, Department of Physics, Stockholm University, AlbaNova, 10691 Stockholm, Sweden
3. School of Natural Sciences and Medicine, Ilia State University, 0194 Tbilisi, Georgia
4. McWilliams Center for Cosmology and Department of Physics, Carnegie Mellon University, Pittsburgh, PA 15213, USA
* Correspondence: brandenb@nordita.org

**Abstract:** Magnetic helicity plays a tremendously important role when it is different from zero on average. Most notably, it leads to the phenomenon of an inverse cascade. Here, we consider decaying magnetohydrodynamic (MHD) turbulence as well as some less common examples of magnetic evolution under the Hall effect and ambipolar diffusion, as well as cases in which the magnetic field evolution is constrained by the presence of an asymmetry in the number density of chiral fermions, whose spin is systematically either aligned or anti-aligned with its momentum. In all those cases, there is a new conserved quantity: the Hosking integral. We present quantitative scaling results for the magnetic integral scale as well as the magnetic energy density and its spectrum. We also compare with cases were a magnetic version of the Saffman integral is initially finite. Rotation in MHD turbulence tends to suppress nonlinearity and thereby also inverse cascading. Finally, the role of the Hosking and magnetic Saffman integrals in shell models of turbulence is examined.

**Keywords:** decaying turbulence; MHD turbulence; chiral magnetic effect; Hall effect; shell models

## 1. Introduction

This paper is part of a special issue commemorating the work of Jack Herring. His scientific career started off with papers on the effect of the solar wind on the lunar atmosphere in 1959 [1]. In 1961, he extended this work to exoplanet atmospheres [2]. He also worked on stellar opacities [3]. In all those cases, he was very much ahead of its time. At the time of Parker's prize-winning paper on the discovery of non-static solutions [4], the physical reality and properties of the solar wind were still rather unclear and under-appreciated. Likewise, Herring's work on stellar opacities was well before proper numerical stellar structure and evolution models became available; the Henyey method [5] (solving a matrix equation instead of using an iterative shooting method from both ends) became known only in 1964. Subsequently, Herring turned to hydrodynamic convection and turbulence – topics that then determined much of his future work. During his career, he never really worked on magnetic fields or helicity, but he did interact with people on a daily basis, who were very much involved in these subjects, both early on [6] and also later during his career [7]. It is therefore not surprising that this special issue also extends to topics involving magnetic fields and helicity.

Having helicity in a system usually requires external factors such as stratification and rotation [8–10]. In this sense, the absence of helicity may be regarded as the more generic situation. It may therefore also seem natural that helicity does not play an important role when it is absent on average. This is believed to be the case in hydrodynamic turbulence, but it changes when magnetic fields are involved. Although both kinetic and magnetic helicities are ideal invariants, only the magnetic helicity has a non-ideal dissipation that is slower than that of the magnetic energy. By contrast, the dissipation of kinetic helicity is

faster than that of kinetic energy [11,12]. Therefore, in the magnetic case, helicity plays a very important role in a way that is unknown in the hydrodynamic context. But is this still true when the net magnetic helicity is actually zero?

The physical situations of interest here include the decay of primordial magnetic fields in the early Universe during the radiation-dominated era, when the electric conductivity is high and the initially generated magnetic field can only decay. When the plasma is hot enough, the chirality of fermions also plays an important role, leading to an interplay with magnetic helicity. Another situation of interest is when only the Hall effect plays a role, so there are then no fluid motions, but just the flow of electrons. This is relevant in neutron star crusts, which are solid, so the ions are immobile. Another application is to the solar wind on scales below the proton gyroscale, where the ions create a smooth motionless background. The induction equation with just the Hall effect included leads to interesting decay dynamics—remarkably similar to ordinary magnetohydrodynamics (MHD). In all those cases, magnetic helicity can play a role even when it vanishes on average. In those cases, magnetic helicity fluctuations may be responsible for driving an inverse cascade similar to the case of nonvanishing mean magnetic helicity.

Less obvious examples of the dynamics discussed above include the Sun, because here the magnetic helicity is usually nonvanishing on average [13]. Even in the solar wind, where the conditions resemble those of decaying turbulence, the magnetic helicity is observed to be nonvanishing on average and systematically of opposite signs in the northern and southern hemispheres [14]. Near the ecliptic, however, the magnetic helicity fluctuates around zero [11] and may also be in a state of decay, so this may be another example where magnetic helicity fluctuations play an important role.

## 2. Nonhelical Turbulence and the Hosking Integral

In this section, we discuss the Hosking integral and why it is crucial to understanding nonhelical MHD turbulence with strong magnetic fields. Unlike the case of weak magnetic fields, when the dynamics is still controlled by the presence of hydrodynamic effects, we are dealing here with effects that are specific to the presence of magnetic fields, albeit with zero average. We focus on decaying turbulence.

### 2.1. Nonhelical Inverse Cascading and Scaling Relations

Already in 2001, it was noted that, even in the nonhelical case of a turbulently decaying magnetic field, there is a small amount of inverse cascading in the sense that for wavenumbers below the peak, the magnetic energy spectrum rises with time uniformly for all lower $k$ [15]. The actual amount of this rise was small and one could have argued that it was just because of numerical inaccuracies. Subsequent simulations [16], however, confirmed such inverse cascading and those authors discussed the potential interplay between the shallower kinetic energy spectrum proportional to $k^2$ and the steeper magnetic energy spectrum proportional to $k^4$. The qualitative idea was that the shallower velocity spectrum pushes the magnetic spectrum upward, which then would drive more kinetic energy at small $k$, and so forth.

The choice of the initial magnetic energy being proportional to $k^4$ is important here. When such a spectrum was used in the first numerical simulations [15], the authors made reference to the early work in Ref. [17], where causality arguments were put forward. Nowadays, however, Ref. [18] has become the standard reference for the choice of an initial $k^4$ spectrum. Later, it turned out that with a shallower initial $k^2$ spectrum, no inverse cascading can be found [19,20]. The reason for this particular aspect will be discussed in more detail in this paper.

In 2014, the idea of an inverse cascade in the nonhelical case with a $k^4$ spectrum became really very clear [21]. This paper was on the arXiv since April 2014, but the paper was published only in February 2015. The results were reproduced in the relativistic context in Ref. [22]. Their work was on the arXiv since July 2014 and made reference to the 2015 paper. The significance of both findings is that it presents early support for the

subsequent discovery of the Hosking integral as a new invariant in MHD turbulence at large magnetic Reynolds numbers.

When the Hosking integral was discovered in Ref. [23], it was originally called the "Saffman helicity invariant". As already pointed out in Ref. [24], H. K. Moffatt informed the community of the fact that this term may be misleading, because the term 'helicity invariant' is reserved for integrals that are chiral in character. He also recalled that Saffman never considered helicity in his papers. The term "magnetic helicity density correlation integral" may be more appropriate, but it is rather clumsy. Following [25], where this quantity was called the Hosking integral, this term continued being used by others [26,27]. It should also be noted that 'integral' instead of 'invariant' is appropriate since applications to turbulence apply always to finite Reynolds and Lundquist numbers. In this connection, it should be emphasized that the Hosking integral tends to decay with time in a power-law fashion and that the exponent decreases with increasing Lundquist number Lu approximately as $\mathrm{Lu}^{-1/4}$ [24].

The energy decay in turbulence is usually characterized by the energy spectrum $E(k, t)$. In the following, we sometimes add the subscripts K and M for kinetic and magnetic energy spectra and other quantities. We focus here on magnetic energy spectra, $E_\mathrm{M}(k, t)$, which are defined such that $\int E_\mathrm{M}(k, t)\, \mathrm{d}k = \langle \boldsymbol{B}^2 \rangle / 2\mu_0 \equiv \mathcal{E}_\mathrm{M}(t)$ is the magnetic energy, and $\mu_0$ is the vacuum permeability. The decay can then be parameterized by $\mathcal{E}_\mathrm{M}(t)$ and the magnetic integral scale, which is defined in terms of the magnetic energy spectrum as

$$\xi_\mathrm{M}(t) = \int_0^\infty k^{-1} E_\mathrm{M}(k, t)\, dk \bigg/ \int_0^\infty E_\mathrm{M}(k, t)\, dk. \tag{1}$$

One can always attempt to describe the relations for $\xi_\mathrm{M}(t)$ and $\mathcal{E}_\mathrm{M}(t)$ through power laws. In addition, the spectrum can evolve underneath an envelope,

$$E_\mathrm{M}(k, t) \leq \mathrm{const} \times k^\beta, \tag{2}$$

which is in general different from the initial subinertial range spectrum, $E_\mathrm{M}(k, t_0) = \mathrm{const} \times k^\alpha$, where $\alpha$ is the subinertial range slope. The three relations for $\xi_\mathrm{M}(t)$, $\mathcal{E}_\mathrm{M}(t)$, and $E_\mathrm{M}(k, t)$ can then be constrained through dimensional arguments once we have a good idea about the relevant dimensional quantity that governs the decay.

In 2017, the decay of a nonhelical turbulent magnetic field is found to be described by an exponent $\beta$ that was determined to be between $\beta = 1$ [28] and $\beta = 2$ [20], but it was unclear why any of those two possibilities, or any other one, would have to be expected. This is what the Hosking integral now explains, namely that $\beta = 3/2$.

Figure 1a shows magnetic energy spectra at four different times for a nonhelical magnetically dominated run corresponding to Run K60D1bc in Ref. [24]. Here, $k$ is normalized by the initial peak wavenumber $k_0$. We clearly see that the spectrum exhibits inverse cascading in that the spectral magnetic energy *increases* with time at small $k$, as indicated by the upward arrow on the left. The overall energy does of course decay, as indicated by the decline of the spectral peak and the decrease of spectral energy at large $k$, as indicated by the downward arrow on the right.

To quantify the temporal changes of $\xi_\mathrm{M}(t)$ and $\mathcal{E}_\mathrm{M}(t)$, it is convenient to compute the instantaneous scaling exponents [28]

$$q(t) = \mathrm{d}\ln\xi_\mathrm{M}/\mathrm{d}\ln t \quad \text{and} \quad p(t) = -\mathrm{d}\ln\mathcal{E}_\mathrm{M}/\mathrm{d}\ln t; \tag{3}$$

see Figure 1b. We see that with time (larger red symbols), the solution evolves toward the point $(q, p) = (4/9, 10/9)$, as is also theoretically expected [24]. Although we mainly focus on the case of nonhelical magnetic fields, we also compare in Figure 1b with the expected solution for the fully helical case (orange), and include solutions for hydrodynamic turbulence that are governed either by the Loitsyansky or the Saffman integrals.

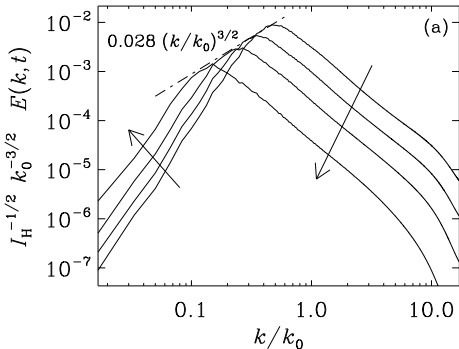 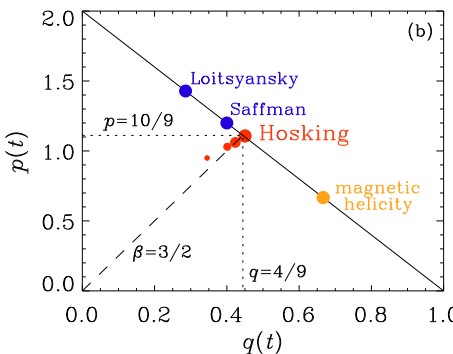

**Figure 1.** (**a**) magnetic energy spectra, normalized by $I_{\rm H}^{-1/2} k_0^{-3/2}$; the dashed dotted line shows the envelope $0.028 (k/k_0)^{3/2}$ under which the spectrum evolves. The times are $ck_1 t = 3, 7, 17,$ and $58$. (**b**) $qp$ diagram showing as red dots the convergence of $p(t)$ versus $q(t)$ toward the Hosking attractor $(q, p) = (4/9, 10/9)$. The blue symbols denote the Loitsyansky and Saffman attractors, respectively, and the orange symbol denotes the magnetic helicity attractor.

Before we continue, it is useful to clarify the concept of what we often refer to as a 'governing quantity'. Take, for example, standard hydrodynamic Kolmogorov turbulence. Here, the rate of energy transfer per unit mass $\epsilon$ (which is the rate of energy input and also the rate of energy dissipation) is such a quantity and the relevant physical scaling laws can be expressed in terms of powers of $\epsilon$ and other relevant variables such as the wavenumber $k$ itself. This then yields for the energy spectrum per unit mass the expression $E(k) = C_{\rm K} \epsilon^{2/3} k^{-5/3}$, where $C_{\rm K}$ is a dimensionless coefficient of order unity (the Kolmogorov constant; typically $C_{\rm K} \approx 1.6$). Other such governing quantities include the mean magnetic helicity density $\langle h \rangle$ and some other quantities that are crucial to the physics. They are usually constant or well conserved.

### 2.2. The Loitsyansky and Saffman Integrals in Hydrodynamics

In the hydrodynamic case, the decay of turbulence can follow different behaviors depending on the relevant conservation law. (In practice, conserved quantities are usually not perfectly conserved under turbulent conditions, and some are better conserved than others. Which one is the most relevant quantity depends on the relative conservation properties under different circumstances.) One such conserved quantity is the Loitsyansky integral [29,30],

$$I_{\rm L} = - \int \langle \boldsymbol{u}(\boldsymbol{x}) \cdot \boldsymbol{u}(\boldsymbol{x} + \boldsymbol{r}) \rangle \, r^2 \, {\rm d}^3 \boldsymbol{r}, \tag{4}$$

which is believed to play an important role. This integral reflects the local conservation of angular momentum and has dimensions $[I_{\rm L}] = {\rm m}^7\,{\rm s}^{-2}$. If this quantity governs the decay of turbulence, the time dependence of the growth of the integral scale can be motivated by dimensional arguments as $\xi(t) \propto I_{\rm L}^a t^b$, where the exponents $a$ and $b$ must be, on dimensional grounds, $a = 1/7$ and $b \equiv q = 2/7$. The kinetic energy then obeys $\mathcal{E}_{\rm K} \propto I_{\rm L}^{2/7} t^{-10/7}$, i.e., $p = 10/7$. The envelope under which the peak of the spectrum evolves obeys $E_{\rm K}(k, t) \leq C_{\rm L} I_{\rm L} k^4$.

Another conserved quantity is the Saffman integral,

$$I_{\rm S} = \int \langle \boldsymbol{u}(\boldsymbol{x}) \cdot \boldsymbol{u}(\boldsymbol{x} + \boldsymbol{r}) \rangle \, {\rm d}^3 \boldsymbol{r}, \tag{5}$$

which has dimensions $[I_{\rm S}] = {\rm m}^5\,{\rm s}^{-2}$. Similarly, if this quantity governs the decay of turbulence, the time dependence of $\xi$ must be $\xi(t) \propto I_{\rm S}^a t^b$, where $a = 1/5$ and $b \equiv q = 2/5$ on dimensional grounds. The kinetic energy then obeys $\mathcal{E}_{\rm K} \propto I_{\rm L}^{2/5} t^{-6/5}$, i.e., $p = 6/5$. The envelope under which the peak of the spectrum evolves obeys in this case $E_{\rm K}(k, t) \leq C_{\rm S} I_{\rm S} k^2$.

Whether $I_L$ or $I_S$ determine the decay depends on the existence of long-range correlations, as can be seen from the Taylor expansion of the kinetic energy spectrum as [23,29]

$$2E_K(k \to 0) \equiv \mathrm{Sp}(\boldsymbol{u})(k \to 0) = \frac{I_S}{2\pi^2}k^2 + \frac{I_L}{12\pi^2}k^4 + \ldots, \tag{6}$$

where an initially non-vanishing Saffman integral automatically implies a $k^2$ scaling in the subinertial range. Thus, the decay does depend on the infrared part of the initial kinetic energy spectrum. In that case, the slope is the same as that required for the initial spectrum so that the Saffman integral is indeed nonvanishing. Furthermore, as pointed out in Ref. [23], owing to the invariance of $I_S$ and $I_L$, both an initial $k^2$ and a $k^4$ spectrum will remain unchanged. This implies that there can be no inverse cascading in hydrodynamics.

*2.3. The Magnetic Saffman Integral: Comparison with the Hosking Integral*

As already pointed out in Ref. [23], the formulation of Section 2.2 can also be applied to the magnetic field, except that there is no reason for the magnetic version of the Loitsyansky integral to be conserved. The magnetic Saffman integral (hereafter $I_{SM}$), on the other hand, might indeed be conserved. Physically, it would reflect the local conservation of magnetic flux. Again, when $I_{SM}$ is non-vanishing initially, we expect a quadratic magnetic energy spectrum, which would also persist at later times. For a steeper $k^4$ subinertial range magnetic energy spectrum, however, the magnetic Saffman integral must vanish and the Hosking integral is then expected to play a dominant role. It is defined as

$$I_H = \int \langle h(\boldsymbol{x}) h(\boldsymbol{x} + \boldsymbol{r}) \rangle \, \mathrm{d}^3\boldsymbol{r}, \tag{7}$$

where $h = \boldsymbol{A} \cdot \boldsymbol{B}$ is the magnetic helicity density with dimensions $[h] = [B]^2[x]$. In ordinary MHD, we can express the magnetic field as an Alfvén velocity, i.e., we write the magnetic field in Alfvén units, so $[B] = \mathrm{m\,s}^{-1}$. Therefore, $[h] = [x]^3[t]^{-2}$, and thus $[I_H] = [B]^4[x]^5 = [x]^9[t]^{-4}$. If $I_H$ plays a governing role in the decay, we expect therefore $\xi_M(t) \propto I_H^{1/9} t^{4/9}$, $\mathcal{E}_M \propto I_H^{2/9} t^{-10/9}$, and $E_M(k,t) \leq C_H I_H k^{3/2}$.

The Hosking integral is in general expected to be different from zero [23]. This automatically implies a quadratic scaling of the helicity variance spectrum, $\mathrm{Sp}(h)$. Here, $\mathrm{Sp}(h) = \oint_{4\pi} |\tilde{h}|^2 k^2 \mathrm{d}\Omega_k / (2\pi L)^3$ denotes the shell-integrated spectrum, a tilde marks a quantity in Fourier space, and $\Omega_k$ is the solid angle in Fourier space, so that $\int \mathrm{Sp}(h) \, \mathrm{d}k = \langle h^2 \rangle$. The quadratic scaling for a finite Hosking integral follows from the expansion

$$\mathrm{Sp}(h)(k \to 0) = \frac{I_H}{2\pi^2}k^2 + \ldots \tag{8}$$

In three dimension, a quadratic spectrum corresponds to white noise. We also know that the spectrum of a quadratic quantity cannot be more blue than that of white noise [31], so it seems impossible to have a helicity variance spectrum whose subinertial range is steeper than $k^2$.

In Figure 2, we show magnetic energy and magnetic helicity variance spectra for initial spectra of the form

$$E_M(k, t_0) = \mathrm{const} \times \frac{k^\alpha}{1 + (k/k_0)^{\alpha + 5/3}} \propto \begin{cases} k^\alpha & \text{for } k \ll k_0, \\ k^{-5/3} & \text{for } k \gg k_0, \end{cases} \tag{9}$$

for $\alpha = 2$ and $\alpha = 4$. We solve the isothermal compressible MHD equations using the PENCIL CODE [32] with $1024^3$ mesh points. As expected, and as pointed out previously [19,20], there is inverse cascading only for $\alpha = 4$, but not for $\alpha = 2$. Nevertheless, we see that $\mathrm{Sp}(h)$ retains a $k^2$ spectrum at low wavenumbers in both cases. This suggests that the Hosking integral is indeed always conserved; see Figure 2b,d. It may, however, be less dominant than the magnetic Saffman integral.

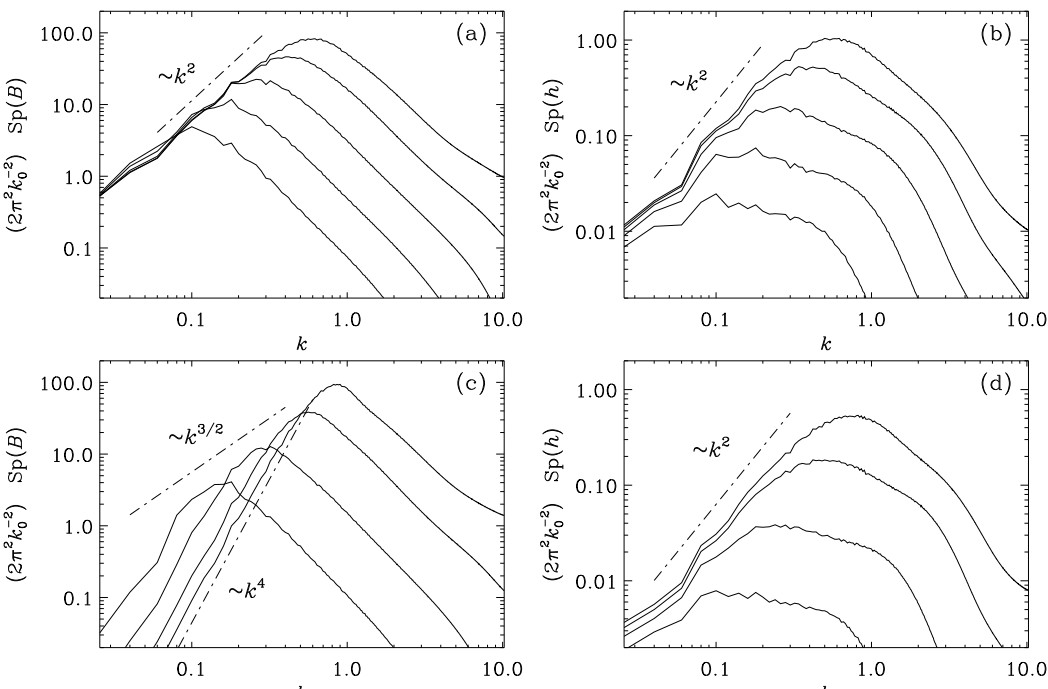

**Figure 2.** Comparison of Sp($B$) (**a,c**) and Sp($h$) (**b,d**) for $\alpha = 2$ (**a,b**) and $\alpha = 4$ (**c,d**). The dashed-dotted lines indicate $k^{3/2}$ scaling in (**c**) and $k^2$ scaling otherwise.

To determine the relevant integrals, $I_H$ and $I_{SM}$, it is convenient to plot compensated spectra. Specifically, to determine $I_{SM}$ and $I_H$, we scale both Sp($B$) and Sp($h$) by $2\pi^2/k^2$. The result is shown in Figure 3. Thus, in summary, we have

$$\xi_M(t) \approx 0.16\, I_{SM}^{1/5} t^{2/5}, \quad \mathcal{E}_M(t) \approx 4.2\, I_{SM}^{2/5} t^{-6/5}, \quad E_M(k) \approx 0.037\, I_{SM}(k/k_0)^2. \tag{10}$$

If the initial spectrum is not $\propto k^2$, but $\propto k^4$, we have

$$\xi_M(t) \approx 0.15\, I_H^{1/9} t^{4/9}, \quad \mathcal{E}_M(t) \approx 3.8\, I_H^{2/9} t^{-10/9}, \quad E_M(k) \approx 0.025\, I_H^{1/2}(k/k_0)^{3/2}. \tag{11}$$

It is remarkable that the prefactors for the Saffman and Hosking scalings are very close to each other; see Table 1 for a summary of the nondimensional prefactors in the relations

$$\xi_M(t) = C_i^{(\xi)} I_i^\sigma t^q, \quad \mathcal{E}_M(t) = C_i^{(\mathcal{E})} I_i^{2\sigma} t^{-p}, \quad E_M(k) = C_i^{(E)} I_i^{(3+\beta)/\sigma}(k/k_0)^\beta, \tag{12}$$

where the index $i$ on the integrals $I_i$ and the coefficients $C_i^{(\xi)}$, $C_i^{(\mathcal{E})}$, and $C_i^{(E)}$ stands for SM or H for magnetic Saffman and Hosking scalings, respectively, and $\sigma$ is the exponent with which length enters in $I_i$: $\sigma = 5$ for the magnetic Saffman integral ($i = SM$) and $\sigma = 9$ for the Hosking integral ($i = H$). Interestingly, while $\beta$ and $p$ can uniquely be related to $q$ via $\beta = 2/q - 3$ and $p = 2(1 - q)$ [28], the exponent $\sigma$ is not uniquely linked to $q$ and we have $\sigma q = 2$ for Saffman scaling and $\sigma q = 4$ for Hosking scaling.

The value of $C_{SM}^{(E)}$ only makes sense when $\alpha = \beta = 2$, while that of $C_H^{(E)}$ only makes sense when $\alpha = 4$ and $\beta = 3/2$. For the other cases, the subinertial range spectrum is not parallel to $k^\beta$, so $\alpha$ and $\beta$ are said to be incompatible with each other (see Table 1) and the given values of $C_{SM}^{(E)}$ and $C_H^{(E)}$ only yield crossings in the middle of the subinertial range.

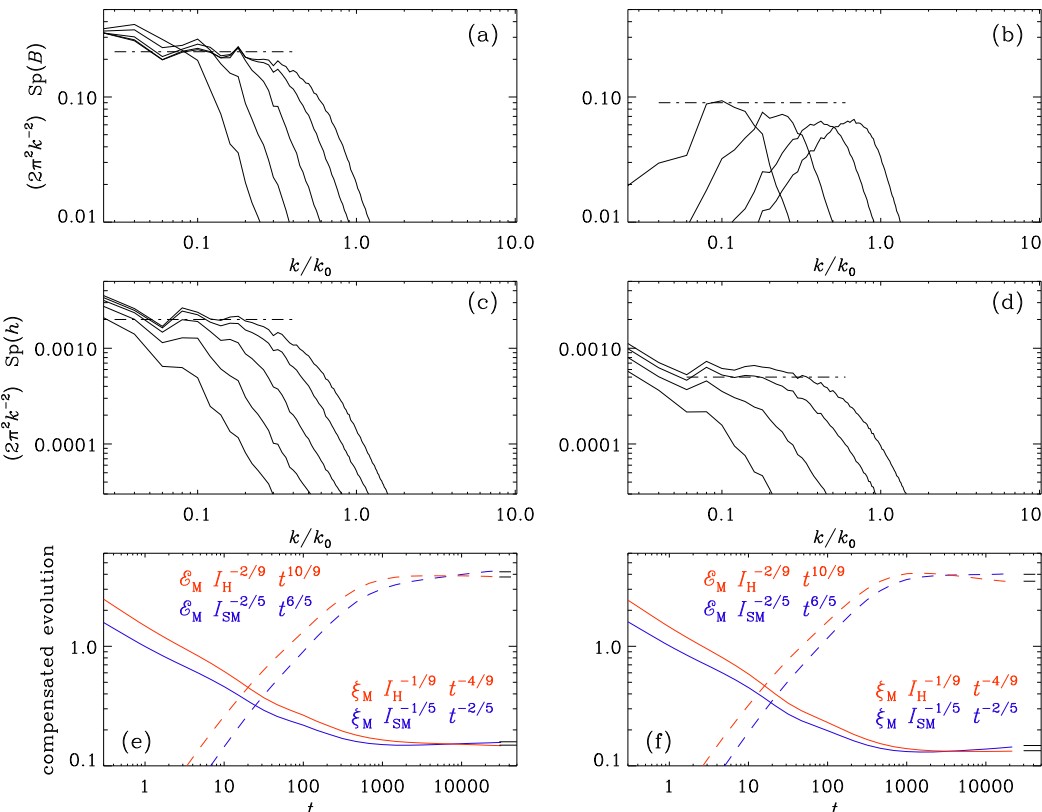

**Figure 3.** Compensated spectra for $\alpha = 2$ (**a,c,e**) and $\alpha = 4$ (**b,d,f**). From (**a,b**), the horizontal dashed-dotted lines indicate that $I_{SM} \approx 0.23$ and $0.09$, respectively, and from (**c,d**) they indicate that $I_H \approx 2 \times 10^{-3}$ and $5 \times 10^{-4}$, respectively. The asymptotic values estimated from (**e,f**) are discussed in the text and in Table 1.

**Table 1.** Summary of nondimensional prefactors in the relations for $\xi_M(t)$, $\mathcal{E}_M(t)$, and $E_M(k,t)$. The numbers in parentheses indicate that the slope $\beta$ is incompatible with the value of $\alpha$.

| $\alpha$ | $\beta$ | $C_{SM}^{(\xi)}$ | $C_H^{(\xi)}$ | $C_{SM}^{(\mathcal{E})}$ | $C_H^{(\mathcal{E})}$ | $C_{SM}^{(E)}$ | $C_H^{(E)}$ |
|---|---|---|---|---|---|---|---|
| 2 | 2 | 0.16 | 0.15 | 4.2 | 3.8 | 0.025 | (0.05) |
| 4 | 3/2 | 0.15 | 0.13 | 4.0 | 3.5 | (0.02) | 0.037 |

We see from Figure 3a that for $\alpha = 2$, the compensated value $(2\pi/k^2)\,\mathrm{Sp}(\boldsymbol{B}) \to I_{SM} \approx 0.2$. For $\alpha = 4$, on the other hand, we only see a flat envelope, i.e., $(2\pi/k^2)\,\mathrm{Sp}(\boldsymbol{B}) \leq 0.1$, i.e., $2E_M(k,t) \leq 0.1/(2\pi^2/k^2)\,(k/k_0)^2$. From Figure 3c,d, we see that $(2\pi/k^2)\,\mathrm{Sp}(h) \to I_H \approx 0.001$ in both cases, i.e., for $\alpha = 2$ and $\alpha = 4$, respectively.

Given that we now know the values of $I_{SM}$ and $I_H$, we can compensate the time evolutions of $\xi_M(t) \propto t^q$ with $q = 2/5 = 0.4$ and $q = 4/9 \approx 0.44$, and those of $\mathcal{E}_M(t) \propto t^{-p}$ with $p = 6/5 = 1.2$ and $p = 10/9 \approx 1.1$. The results for the corresponding coefficients in Equation (12) are summarized in Table 1.

### 2.4. The Effect of Rotation

Rotation suppresses hydrodynamics turbulence. This is modelled by including the Coriolis force, $-2\boldsymbol{\Omega} \times \boldsymbol{u}$, on the right-hand side of the momentum equation, which then reads

$$\frac{D\boldsymbol{u}}{Dt} = -c_s^2 \boldsymbol{\nabla} \ln \rho - 2\boldsymbol{\Omega} \times \boldsymbol{u} + \frac{1}{\rho}[\boldsymbol{J} \times \boldsymbol{B} + \boldsymbol{\nabla} \cdot (2\rho\nu\mathbf{S})], \tag{13}$$

where $D/Dt = \partial/\partial t + \boldsymbol{u} \cdot \boldsymbol{\nabla}$ is the advective derivative, $c_s$ is the isothermal sound speed, $\boldsymbol{\Omega}$ is the angular velocity, $\boldsymbol{J} = \boldsymbol{\nabla} \times \boldsymbol{B}/\mu_0$ is the current density, $\mu_0$ is the permeability, $\rho$ is

the density, $\nu$ is the viscosity, and $\mathsf{S}_{ij} = (\partial_i u_j + \partial_j u_i)/2 - \delta_{ij}\boldsymbol{\nabla}\cdot\boldsymbol{u}/3$ are the components of the rate-of-strain tensor.

In Figure 4, we show $\mathrm{Sp}(\boldsymbol{B})$ and $\mathrm{Sp}(h)$ for $\Omega/c_s k_0 = 10^{-3}$, 0.01, 0.1, and 1 for runs with $\alpha = 4$, which are otherwise the same as that of Figure 2c,d. We see a clear suppression of inverse cascading already for $\Omega/c_s k_0 = 10^{-3}$ and a very strong suppression when this parameter is unity. This is caused by the suppression of the turbulent velocity and thereby of the $\boldsymbol{u}\times\boldsymbol{B}$ nonlinearity in the induction equation.

To express the angular velocity in a physically more meaningful way, we note that in the run with $\Omega/c_s k_0 = 1$, the rms Mach number, $u_{\mathrm{rms}}/c_s$, drops below $10^{-3}$ by the end of the run, while for $\Omega/c_s k_0 = 10^{-3}$, it still stays well above $10^{-3}$. This means that in the latter, the Coriolis number, $\mathrm{Co} \equiv 2\Omega/u_{\mathrm{rms}}k_0$, is around unity when rotational suppression becomes appreciable. At $t = 10^4$, the values of Co are 1.2, 22, 500, and $10^4$ for our four runs in Figure 4.

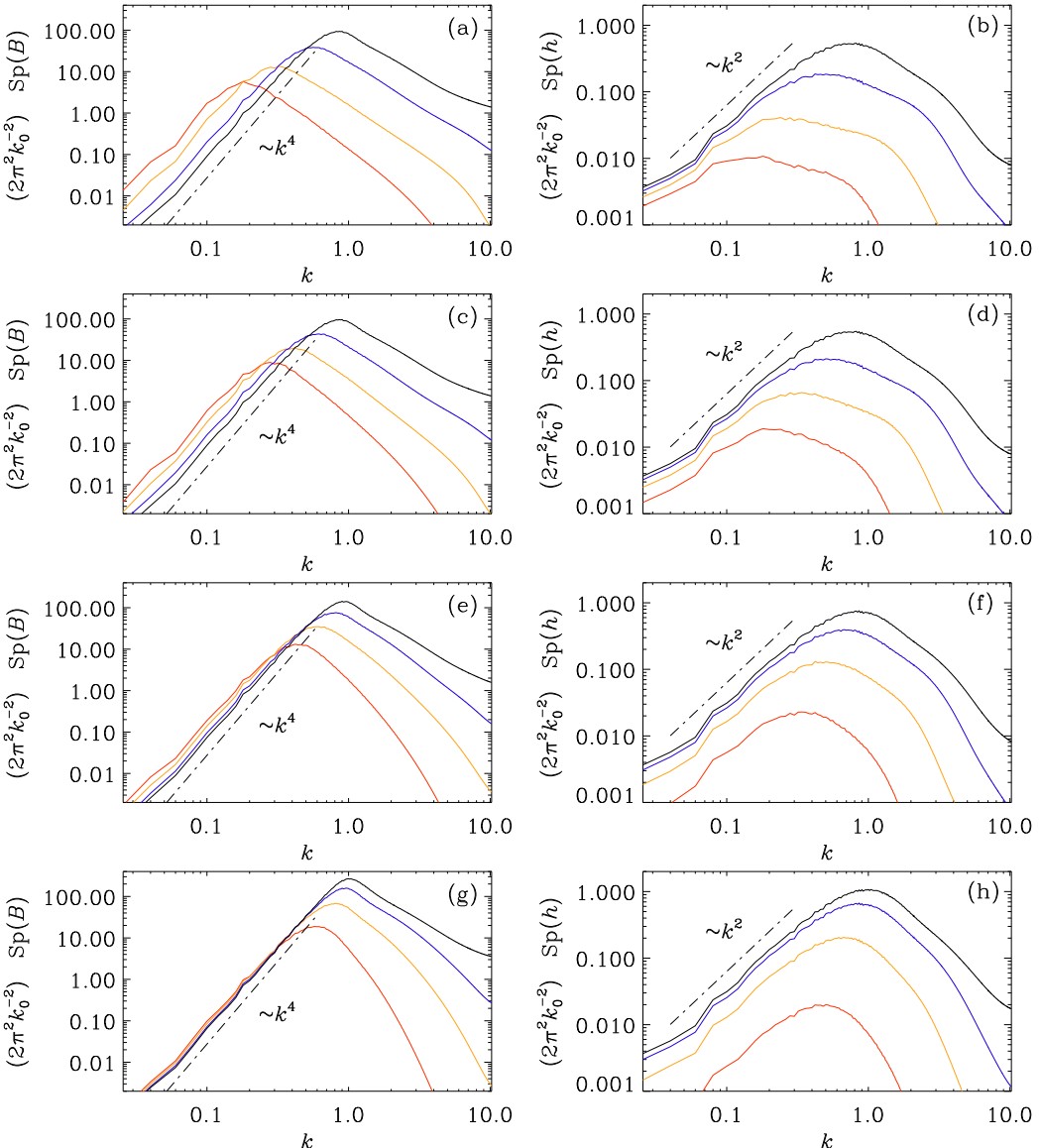

**Figure 4.** $\mathrm{Sp}(\boldsymbol{B})$ (**a,c,e,g**) and $\mathrm{Sp}(h)$ (**b,d,f,h**) for $\Omega/c_s k_0$ from $10^{-3}$ (**a,b**: Co = 1.2) to $\Omega/c_s k_0 = 1$ (**g,h**: Co = $10^4$). The times are 220 (black), 1000 (blue), 4600 (orange), and 22,000 (red).

## 3. Extensions of the Hosking Idea

Equation (7) is the Hosking integral in its original form. In the meantime, two further variants of $I_H$ have been considered. One is where $h$ has been replaced by $h_{\mathrm{tot}} = \boldsymbol{A}\cdot\boldsymbol{B} + 2\mu_5/\lambda$,

where $\mu_5$ is the chiral chemical potential (here in units of an inverse length) and $\lambda$ is a coefficient that quantifies the coupling between fermions and electromagnetic fields. The case $\langle h_{\text{tot}} \rangle = 0$ has been studied recently in Ref. [33]. Another variant of the Hosking integral is that in the case where the magnetic field is controlled by the electromagnetic induction from the Hall effect, which we discuss next.

### 3.1. Hall Effect

In neutron star crusts, the ions are immobile and the current is only carried by electrons with the velocity $\boldsymbol{u}_{\text{e}} = -\boldsymbol{J}/en_{\text{e}}$, where $e$ is the electric charge and $n_{\text{e}}$ is the electron density. As alluded to in the introduction, a similar situation occurs in the solar wind on scales below the proton gyroscale, where the ions constitute a smooth background [34]. The induction equation with the induction from $\boldsymbol{u}_{\text{e}} \times \boldsymbol{B}$ therefore takes the form [35]

$$\frac{\partial \boldsymbol{B}}{\partial t} = \boldsymbol{\nabla} \times \left( -\frac{1}{en_{\text{e}}} \boldsymbol{J} \times \boldsymbol{B} - \eta \mu_0 \boldsymbol{J} \right), \tag{14}$$

where $\eta$ is the magnetic field diffusivity. In the presence of magnetic helicity, one finds an inverse cascade with an overall decay of the magnetic field and a growth of spectral magnetic energy at small wavenumbers below that of the peak of the spectrum [34]. In the present paper, however, the focus is on the nonhelical case, which was already considered in B20, but understood mathematically only later [36].

In this context, it is important to note that the natural dimensions of the magnetic field here are no longer $\text{m s}^{-1}$, but $\text{m}^2\,\text{s}^{-1}$. This was already emphasized in Ref. [37], who used $e = 1.6 \times 10^{-19}\,\text{A s}$, $\mu_0 = 4\pi \times 10^{-7}\,\text{T m A}^{-1}$, and $n_{\text{e}} \approx 2.5 \times 10^{40}\,\text{m}^{-3}$ for neutron star crusts, so we have $en_{\text{e}}\mu_0 \approx 5 \times 10^{15}\,\text{T s m}^{-2}$, and therefore

$$\frac{B}{en_{\text{e}}\mu_0} = \frac{B}{5 \times 10^{15}\,\text{T}}\,\frac{\text{m}^2}{\text{s}}, \tag{15}$$

which is why we say $B$ has dimensions of $\text{m}^2\,\text{s}^{-1}$ in the Hall cascade. This modifies all the dimensional arguments related to $\boldsymbol{B}$ correspondingly. In particular, the units of the magnetic helicity are $[h] = \text{m}^5\,\text{s}^{-2}$ and those of energy spectra are also $\text{m}^5\,\text{s}^{-2}$. Therefore, one has $q = p = 2/5$. This scaling was confirmed in Ref. [37].

In the nonhelical case, the modified Hosking integral has dimensions $m^{13}\,s^{-4}$, and therefore $q = 4/13$ (instead of $4/9$ in MHD). Furthermore, $p = 10/13$ (instead of $10/9$ in MHD), but still $\beta = 3/2$ (just like in MHD). While such a scaling was already seen in the original simulations of Ref. [37], the work in Ref. [36] showed that the modified Hosking integral is indeed conserved. In Figure 5a,b,d,e, we show that, also for Hall dynamics, the Saffman scaling is obeyed for $\alpha = 2$, while Hosking scaling is obeyed for $\alpha = 4$.

It should be noted that earlier work on the Hall cascade focussed on the concept of Whistler turbulence [38], where the Whistler time $t_{\text{w}}$ was identified as the governing timescale [34]. The definition of $t_{\text{w}}$ in Ref. [34] involved the electron plasma frequency and the electron gyrofrequency such that the electron mass drops out. Therefore, $t_{\text{w}}$ can more easily be written as

$$t_{\text{w}} = \frac{L^2}{B_{\text{rms}}/en_{\text{e}}\mu_0}, \tag{16}$$

which is just the characteristic time based on the magnetic field expressed in units of a diffusivity; see Equation (15). Earlier interpretations in terms of Whistler waves [34] seem therefore artificial and obscured the relevant interpretation of the magnetic field as a quantity with dimensions of $\text{m}^2\,\text{s}^{-1}$.

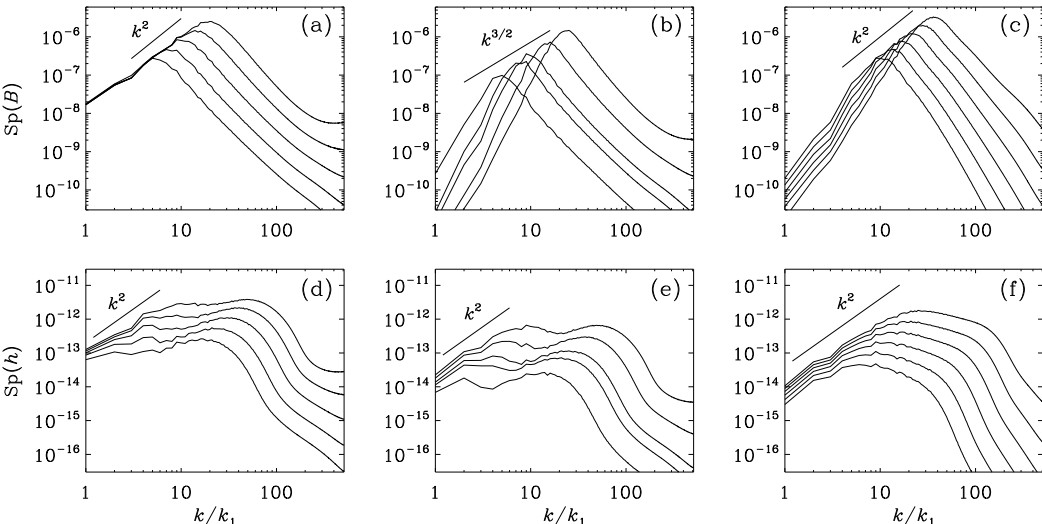

**Figure 5.** Sp($B$) (**a**–**c**) and Sp($h$) (**d**–**f**) for Hall dynamics with $\alpha = 2$ (**a**,**d**) and $\alpha = 4$ (**b**,**e**), and for ambipolar diffusion with $\alpha = 4$ (**c**,**f**). Note the presence of inverse cascading for $\alpha = 4$ in panels (**b**,**c**), although Sp($h$) changes at $k/k_0 \ll 1$ in all cases. The straight lines indicate $k^{3/2}$ scaling in (**b**) and $k^2$ scaling otherwise.

Observationally, the inverse cascade with helicity may be more easily accessible, and this has already been tried in the context of neutron stars [39]. Measuring inverse cascading in the solar wind is conceptually harder, because the system is statistically steady and more advanced stages of decay would only correspond to larger distances from the Sun. On the other hand, most of the spacecrafts are located near the ecliptic and might therefore allow better insight into nonhelical inverse cascading. Unfortunately, most of the observations focus on the high frequency range of the spectrum [40], while the low frequency range is dominated by noise, making it nearly impossible to say anything about inverse cascading to even larger scales.

### 3.2. Ambipolar Diffusion

The Hall effect is a two-fluid effect where the two components are the positive and negative charge carriers. Another two-fluid effect is ambipolar diffusion where the charged fluid with positive and negative charge carriers is taken as one component and neutrals are taken as the other component. The governing equation is

$$\frac{\partial \boldsymbol{B}}{\partial t} = \boldsymbol{\nabla} \times \left( -\frac{\boldsymbol{J} \times \boldsymbol{B}}{\rho_i \nu_{\text{in}}} \times \boldsymbol{B} - \eta \mu_0 \boldsymbol{J} \right), \tag{17}$$

where $\rho_i$ is the ion density and $\nu_{\text{in}}$ is the ion–neutral collision frequency.

Unlike the Hall effect in neutron star crusts, where the magnetic field is said to have dimensions of $\text{m}^2\,\text{s}^{-1}$, we can here write

$$\frac{B}{\sqrt{\rho_i \mu_0}} = \frac{B}{1.5 \times 10^{-16}\,\text{T}}\,\frac{\text{m}}{\text{s}}, \tag{18}$$

where we used $\rho_i = 1.7 \times 10^{-26}\,\text{kg}\,\text{m}^{-3}$ for the interstellar medium with an ionization fraction of $10^{-5}$ and a neutral density of one proton per cubic centimeter. This is why we say that with ambipolar diffusion, just like in MHD, $B$ has dimensions of $\text{m}\,\text{s}^{-1}$. For this reason, we also see in Figure 5c,f qualitatively the same decay behavior as in ordinary MHD.

### 3.3. Chiral MHD

For chiral MHD, the induction equation attains an extra term under the curl that leads to a contribution to the electric field proportional to the product of the magnetic field and a pseudoscalar given by the chiral chemical potential, expressed here as a wavenumber [41].

$$\mu_5 = 24\,\alpha_{\mathrm{em}}\,(n_{\mathrm{L}} - n_{\mathrm{R}})\,(\hbar c/k_{\mathrm{B}}T)^2, \tag{19}$$

where $\alpha_{\mathrm{em}} \approx 1/137$ is the fine structure constant, and $n_{\mathrm{L}}$ and $n_{\mathrm{R}}$ are the number densities of left- and right-handed fermions, respectively. The uncurled induction equation takes then the form

$$\frac{\partial \boldsymbol{A}}{\partial t} = \eta\,(\mu_5 \boldsymbol{B} - \mu_0 \boldsymbol{J}) + \boldsymbol{u} \times \boldsymbol{B}, \quad \boldsymbol{J} = \boldsymbol{\nabla} \times \boldsymbol{B}/\mu_0. \tag{20}$$

The term $\eta\mu_5\boldsymbol{B}$ leads to a growth of the magnetic field for wavenumbers $k < \mu_5$, just in the same way as in mean-field dynamo theory [8–10], but here no mean-field theory is invoked. The generated magnetic field is fully helical, but the relevant quantity is now the *total* chirality density

$$h_{\mathrm{tot}} = \boldsymbol{A} \cdot \boldsymbol{B} + 2\mu_5/\lambda, \tag{21}$$

and it is its volume average that is conserved, i.e., $\langle h_{\mathrm{tot}} \rangle = \mathrm{const}$, provided the boundary conditions are periodic and/or closed, i.e., perfectly conducting. As the magnetic field grows, $\mu_5$ decreases. The rate of this change is proportional to the parameter $\lambda$, which we take here as an adjustable parameter, but in reality is it given by an expression involving the temperature; see Equation (49) of Ref. [41].

It is important to point out that the *physical* chiral chemical potential (which has the units of an energy) is sometimes defined differently. First, the authors of Ref. [33] used Lorentz-Heaviside units, which implies another factor of $4\pi$ in the numerator of the conversion factor (or rather the lack of a $4\pi$ factor in the denominator), and, second, there is also a factor of 2 in the denominator, so $\hbar c/8\alpha_{\mathrm{em}}$ instead of $\hbar c/4\alpha_{\mathrm{em}}$ for the conversion factor of [33], because they defined their physical chiral chemical potential as half the difference between the physical right- and left-handed chiral chemical potentials. In addition, there is a sign difference between Refs. [41] and [33], but this affects only the physical chiral chemical potential and not our equations, where $\mu_5$ has the units of a wavenumber.

It turns out, perhaps not surprisingly, that in this case, when $\langle h_{\mathrm{tot}} \rangle = 0$, the turbulence decays again in such a way that $q = 4/9$ and $p = 10/9$ and, again, $\beta = 3/2$. This is just like in ordinary (but nonhelical) MHD. In this case, however, $\langle h \rangle \neq \mathrm{const}$, but its modulus decays $\propto t^{-r}$ in a way that is compatible with the real-space realizability condition, $|\langle h \rangle| \leq 2\mathcal{E}_{\mathrm{M}}\xi_{\mathrm{M}}$, i.e., $r = p - q = (10 - 4)/9 = 2/3$. This was also confirmed in Ref. [33]. This study was then applied to the problem of baryogenesis [42], where one tries to explain the small excess of matter over antimatter in the Universe, which is referred to as baryon asymmetry.

The Hosking scaling was confirmed for $\langle \mu_5 \rangle \xi_{\mathrm{M}} \gg 1$ [42], but in the opposite limit of $\langle \mu_5 \rangle \xi_{\mathrm{M}} \ll 1$ the Hosking scaling was no longer obeyed and then both $\langle \mu_5 \rangle$ and $\xi_{\mathrm{M}}$ are believed to be approximately independently conserved [33]. Trying to understand this more thoroughly must be a goal for future studies, where one may hope to reach much larger scale separation between the different relevant wavenumbers in the system, such as the wavenumber $k_0$ of the peak of the magnetic energy spectrum and the value of $\mu_5$.

In Figure 6, we plot magnetic energy and magnetic helicity spectra, as well as magnetic helicity variance spectra for a chiral MHD run with balanced chirality and an initial $k^4$ spectrum for the magnetic field. We see standard inverse cascading with $\beta = 3/2$. Next we compare with the case of an initial $k^2$ spectrum; see Figure 7. In this case, there is still weak inverse cascading, which is probably a consequence of the strong contribution from mean magnetic helicity conservation over extended spatial patches.

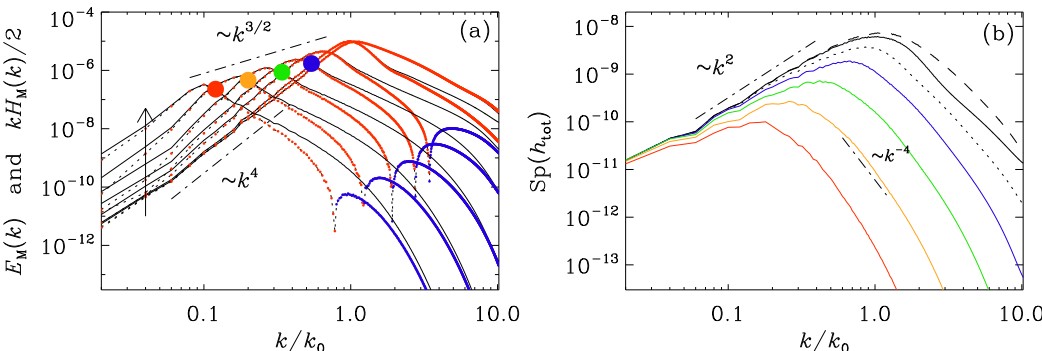

**Figure 6.** (**a**) Magnetic energy (solid lines) and magnetic helicity spectra (dotted lines), and (**b**) magnetic helicity variance spectra for a chiral MHD run with balanced chirality and an initial $k^4$ spectrum for the magnetic field. In (**a**), positive (negative) magnetic helicities are indicated by small red (blue) dots. The four large dots denote the positions of $\xi_M^{-1}$. Their colors are the same as those of the solid lines in (**b**) and correspond to the times 1500, 5000, 15,000, and 50,000.

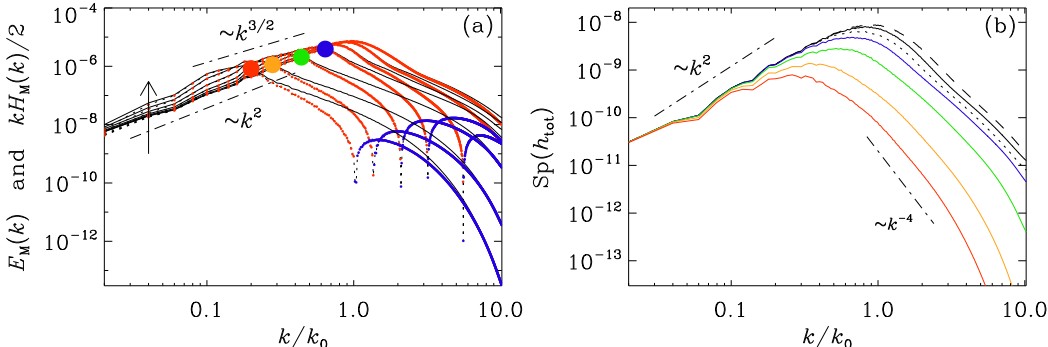

**Figure 7.** Similar to Figure 6, but with an initial $k^2$ spectrum. Note the presence of slight inverse cascading in (**a**), although $\mathrm{Sp}(h_{\mathrm{tot}}) = $ const at $k/k_0 \ll 1$ in (**b**). The different colors refer to the same times as in Figure 6.

Departures from the conservation of the Hosking integral based on $h_{\mathrm{tot}}$ have been seen when $\mu_5 < k_0$ [42], but here we have $\mu_5 > k_0$. To understand more thoroughly the regime where $\mu_5 \gg k_0$, we would need to have much larger numerical resolution. A possible alternative is to use shell models [43], as will be discussed next. However, it is unclear whether such models can capture the relevant effects related to the Hosking integral or the chiral magnetic effect.

## 4. Hosking Integral in Shell Models of Chiral MHD

Shell models describe turbulence through real or complex scalar variables on concentric shells in wavenumber space such that certain conservation laws are obeyed. In MHD, the relevant conservation laws are those of total chirality, total (magnetic plus kinetic) energy, and cross helicity. The Hosking integral describes helicity fluctuations over different scales, and does not have a direct counterpart at the level of shell models. However, the scaling properties resulting from its conservation, could still be manifest in shell models describing the decay of MHD turbulence.

The Hosking integral is particularly important in cases where the mean chirality vanishes. It is also conserved otherwise when the mean total chirality is non-vanishing, but then the conservation of the mean chirality is usually more important. It is also important that the magnetic field is strong, because otherwise the decay properties are dominated by the hydrodynamic turbulent decay. Our goal here is to investigate the decay of magnetic fields with vanishing net chirality in chiral MHD using shell models.

In a shell model, we describe the state of the system in shells of logarithmically spaced wavenumbers $k_n = 2^n$, where $n = 0, 1, 2, \ldots, N$ denotes the shell and $N$ is the truncation level. For $N = 30$, for example, we can span ten orders of magnitude in wavenumber. In MHD, one usually considers complex variables $B_n$ and $u_n$ for the magnetic and velocity fields. The mean magnetic and kinetic energy densities are given by

$$\mathcal{E}_{\mathrm{M}} = \tfrac{1}{2} \sum_{n=0}^{N} |B_n|^2 \quad \text{and} \quad \mathcal{E}_{\mathrm{K}} = \tfrac{1}{2} \sum_{n=0}^{N} |u_n|^2. \tag{22}$$

In shell models, the fluid density is constant and therefore not indicated in the definition of the kinetic energy. Also the permeability factor in the magnetic energy has been omitted.

Magnetic helicity is a signed quantity, i.e., it can be positive or negative. How to describe this in a standard shell model is a matter of convention. One approach is to associate even and odd shells with the decomposition into positively and negatively polarized modes of the field. This idea was first developed for the kinetic helicity [44]. This then leads to the definition of the magnetic helicity as [45–47]

$$\mathcal{H}_{\mathrm{M}} = \sum_{n=0}^{N} (-1)^n |B_n|^2 / k_n, \tag{23}$$

which satisfies the realizability condition

$$k_n |\mathcal{H}_{\mathrm{M}}(k_n)| \leq 2E_{\mathrm{M}}(k_n). \tag{24}$$

To preserve the preferential growth of positively (negatively), polarized modes on even (odd) shells, we write

$$\left[ \eta k (k_n - (-1)^n \mu_5) + \frac{\mathrm{d}}{\mathrm{d}t} \right] B_n = \tfrac{1}{6} \mathrm{i} k_n \left[ M(u, B) - M(B, u) \right], \tag{25}$$

where the $\mu_5$ term leads to a growth of $|b_n|^2$ for even (odd) values of $n$ when $\mu_5$ is positive (negative), and $M(x, y)$ is a nonlinear functional, where $x$ and $y$ stand for the full $n$-dependent arrays. The essence of shell models is to couple only nearest and next-nearest neighbors. We refer to this model as type I. This prescription then leads to

$$M(x, y) = x_{n+1} y_{n+2} + x_{n-1} y_{n+1} + x_{n-2} y_{n-1} \quad \text{(type I)}. \tag{26}$$

As already emphasized in Section 2.4, the velocity plays a crucial role in producing an inverse cascade. It is governed by the Navier-Stokes equation with the Lorentz force included. There are then two further quadratic nonlinearities for $u$ and $B$; see Refs. [45–47] for details.

Another approach to treat helicity is to write the equations separately for the positively and negatively polarized modes and thus have evolution equations for $u_n^{\pm}$ and $B_n^{\pm}$. We refer to this model as type II. The helicity density can then be written as [48]

$$\mathcal{H}_{\mathrm{M}} = \sum_{n=0}^{N} \left( |B_n^{+}|^2 - |B_n^{-}|^2 \right) \Big/ k_n, \tag{27}$$

and the magnetic energy is $\mathcal{E}_{\mathrm{M}} = \sum_{n=0}^{N} (|B_n^{+}|^2 + |B_n^{-}|^2)$. The evolution equations for $B_n^{\pm}$ take then the form

$$\left[\eta k(k_n \mp \mu_5) + \frac{\mathrm{d}}{\mathrm{d}t}\right]B_n^\pm = \tfrac{1}{6}\mathrm{i}k_n\left[M_\pm(u,B) - M_\pm(B,u)\right] \quad \text{(type II)}, \tag{28}$$

where [48]

$$M_\pm(x,y) = x_{n+1}^\mp y_{n+2}^\pm + x_{n-1}^\mp y_{n+1}^\mp + x_{n-2}^\pm y_{n-1}^\mp. \tag{29}$$

Note that for the intermediate terms, the signs in the superscripts are the same, i.e., $u_{n-1}^- B_{n+1}^-$ appear in the evolution of $B_n^+$ and $u_{n-1}^+ B_{n+1}^+$ in the evolution of $B_n^-$; see also Ref. [49], where such models were proposed independently.

In Figure 8, we present models of types I and II with $N = 30$ shells using $\lambda = 10^{10}$, $k_0 = 2^{14} = 16{,}384 \approx 1.6 \times 10^4$, $\nu = \eta = 5 \times 10^{-11}$, and $\mu_5$ is computed as $\mu_5 = -\mu_\mathrm{M}$, where $\mu_\mathrm{M} \equiv \mathcal{H}_\mathrm{M}\lambda/2 \approx 1.8 \times 10^5$ is the chiral chemical potential equivalent of the magnetic helicity.

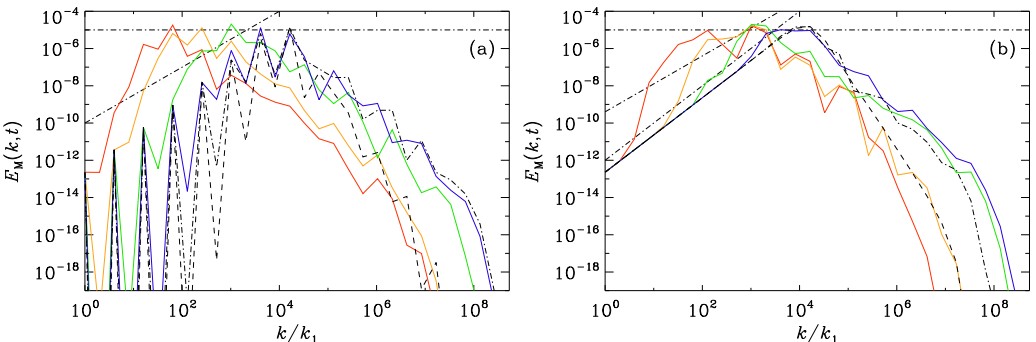

**Figure 8.** Evolution of $E_\mathrm{M}(k,t)$ from shell models of (**a**) type I and (**b**) type II. The times are 10 (red), 1 (orange), 0.1 (green), 0.01 (blue), and earlier times are denoted by black lines of different line types. Note the presence of inverse cascading in both cases.

In all cases, we start with a $k^2$ spectrum, so we expect to see no inverse cascading. Looking at the results of Figure 8, however, this does not seem to be the case. Our results are still preliminary, but our conclusion so far is that shell models may not capture the same inverse cascade behavior that we have found in the direct numerical simulations. On the other hand, more parameter studies are warranted before one can draw more firm conclusions. One must also remember that departures from the conservation of the Hosking integral have been seen in certain direct numerical simulations [42].

## 5. Conclusions

In this paper, we have presented a discussion of the Hosking integral in various contexts in which it has been considered so far: ordinary MHD, MHD with chiral fermions, as well as just the induction equation – either with Hall nonlinearity or with ambipolar diffusion nonlinearity. When the total chirality vanishes (non-chiral case with zero magnetic helicity or chiral case with finite magnetic helicity balancing the fermion chirality) it is the correspondingly adapted Hosking integral that governs the decay of $\mathcal{E}_\mathrm{M}(t) \propto t^{-p}$ and the increase of $\xi_\mathrm{M}(t) \propto t^q$ with $p = 10/9$ and $q = 4/9$ for both ordinary MHD and also just the induction equation with ambipolar diffusion. When the nonlinearity is given by the Hall effect, on the other hand, we have $p = 10/13$ and $q = 4/13$. The case with chiral fermions is somewhat special, because now the magnetic field is actually fully helical, but this helicity is balanced by fermion chirality. Again, in that case the Hosking integral determines the decay behavior. However, there is also another decaying quantity: the mean magnetic helicity density, which is now actually finite and balanced by fermion chirality. It is found to decay like $t^{-2/3}$.

In previous work on decaying turbulence, the decay properties of hydrodynamic and MHD turbulence were motivated by the use of self-similarity and invariance of the governing equations under rescaling [28,50]. This is different in the present work where we have just made use of dimensional arguments. Still, the use of invariance under rescaling

is necessary to motivate the equilibrium line $p = 2(1 - q)$ in the $qp$ diagram in Figure 1b. It will therefore be interesting to find out whether the existence of this line could also be motivated by other means. It probably can, as implied by the derivation of Equation (12), and this might reveal a more basic relation to the parameter that there was called $\sigma$.

An open question is whether the Hosking integral can also play a role in driven MHD turbulence, for example. One possibility could be the production of inverse cascade behavior where magnetic energy grows on wavenumbers below the energy injection wavenumber. This could then leads to a turbulent subinertial range scaling of the form

$$E_{\mathrm{M}}(k) \propto I_{\mathrm{H}}^a k^b. \tag{30}$$

Using dimensional arguments, we would find $3 = 9a - b$ and $2 = 4a$ for balancing the dimensions of length and time, respectively. Therefore, $a = 1/2$ and $b = 3/2$. Thus, $b$ is positive and equal to the Kazantsev slope known in kinematic nonhelical small-scale dynamos [51]. Whether or not there is actually a connection with Kazantsev's small-scale dynamo theory remains another open question.

In our work we have also examined whether some aspects of the Hosking integral might be reproducible with shell models. At the moment, this does not seem to be the case, but this could well be a consequence of not having performed sufficiently extensive parameter studies. Thus, more work might be warranted.

**Author Contributions:** Conceptualization, A.B. and G.L.; methodology, A.B.; software, A.B. and G.L. All authors have read and agreed to the published version of the manuscript.

**Funding:** This research was funded by Vetenskapsrådet grant number 2019-04234 and NASA ATP award number 80NSSC22K0825.

**Data Availability Statement:** The source code used for the simulations of this study, the PENCIL CODE [32], is freely available on https://github.com/pencil-code/ (accessed on 1 May 2023). The DOI of the code is https://doi.org/10.5281/zenodo.2315093 (accessed on 1 May 2023). The simulation setups and the corresponding secondary data are available on http://norlx65.nordita.org/~brandenb/projects/Hosking-Shell (accessed on 1 May 2023).

**Conflicts of Interest:** The authors declare no conflict of interest.

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
