# Peer review of "Turbulence with Magnetic Helicity That Is Absent on Average"

_atmosphere, doi:10.3390/atmos14060932_

Round 1

Reviewer 1 Report

The paper is very interesting indeed and opens a new perspective in the topic. The role of magnetic helicity fluctuations withvanishing mean value is indeed a new direction to be investigated. The paper presents a theoretical perspective for the problem. The presentation is convincing and interesting. The only shortcoming which I see here is that the authors do not mentioned that magnetic helicity is not a theoretical speculation only rather it can be measured say in solar active regions. A reference on such kind of researches would substantially improve the Introduction and/or Discussion.

Author Response

We thank the two anonymous referees for the stimulating comments that
have led to corresponding changes in the manuscript. Those changes are
marked in blue fonts. We often give line numbers, but some lines are
not being correctly numbered by the journal macros.

Referee 1

> The paper is very interesting indeed and opens a new perspective in
> the topic. The role of magnetic helicity fluctuations with vanishing mean
> value is indeed a new direction to be investigated. The paper presents a
> theoretical perspective for the problem. The presentation is convincing
> and interesting. The only shortcoming which I see here is that the
> authors do not mentioned that magnetic helicity is not a theoretical
> speculation only rather it can be measured say in solar active regions.
> A reference on such kind of researches would substantially improve the
> Introduction and/or Discussion.

We have now added a new paragraph where we discuss magnetic helicity
fluctuations in the ecliptic of the solar wind as another potential
example where magnetic helicity fluctuations could explain inverse
cascade behavior; see lines 46-48.

At the end of the penultimate paragraph of the introduction, we have now
also explained that inverse cascading is a manifestation of the presence
of a new conserved quantity; see lines 50-53.

We have added an additional paragraph regarding observations of the Hall
case on page 9, lines 234-242.

Reviewer 2 Report

This paper discusses the properties of decaying MHD turbulence in the presence of magnetic helicity, which is a crucial ingredient in many astrophysical situations. In particular, the authors introduce a relatively new quantity called the Hosking integral. Using direct numerical simulations, they show that it is a conserved quantity that provides a better understanding of MHD turbulence.

The paper can be seen as a review of recent work done by one of the two authors. It is a relevant work as it is always difficult to keep up with the literature on a specific issue. The paper is very well written and accessible to anyone who wants to learn more. I only have minor comments that could help improve the paper slightly and I do not need to read the new version.

Minor comments:

I suggest changing the title to include ‘mhd turbulence’.

The application of the Hall effect is limited to neutron star crust, however, in astrophysics Hall MHD turbulence is often used to describe, for example, solar wind turbulence at sub-ion scales, where in situ measurements are also available. A reference to this topic (with a citation such as a review/book) could be made. 

In Cho (PRL, 2011) - figure 2, I see a behavior that resembles that described here. The interpretation involves whistler wave packets. How can such physics impact the interpretation given here?

A final question concerns additional effect, such as rotation, which is important for the geo-dynamo problem. In principal, is it possible to generalize this invariance to rotating MHD?

Author Response

We thank the two anonymous referees for the stimulating comments that
have led to corresponding changes in the manuscript. Those changes are
marked in blue fonts. We often give line numbers, but some lines are
not being correctly numbered by the journal macros.

Referee 2

> This paper discusses the properties of decaying MHD turbulence in
> the presence of magnetic helicity, which is a crucial ingredient in
> many astrophysical situations. In particular, the authors introduce
> a relatively new quantity called the Hosking integral. Using direct
> numerical simulations, they show that it is a conserved quantity that
> provides a better understanding of MHD turbulence.
> The paper can be seen as a review of recent work done by one of the two
> authors. It is a relevant work as it is always difficult to keep up with
> the literature on a specific issue. The paper is very well written and
> accessible to anyone who wants to learn more. I only have minor comments
> that could help improve the paper slightly and I do not need to read
> the new version.
> Minor comments:
> I suggest changing the title to include "mhd turbulence".

Thanks for the suggestion; including "turbulence" is indeed important,
so have call the paper now:

  Turbulence with Magnetic Helicity that is Absent on Average

> The application of the Hall effect is limited to neutron star crust,
> however, in astrophysics Hall MHD turbulence is often used to describe,
> for example, solar wind turbulence at sub-ion scales, where in situ
> measurements are also available. A reference to this topic (with a
> citation such as a review/book) could be made.

We have now made reference to applications in the solar wind; see
the blue text fragments on pages~2 and 8; see lines 46-48, and also
in the tex just before Eq. (13).

> In Cho (PRL, 2011) - figure 2, I see a behavior that resembles that
> described here. The interpretation involves whistler wave packets. How
> can such physics impact the interpretation given here?

We have now referred to Cho11 in lines 210-214. It was also referred to in
B20, where the focus was mainly on cases with helicity on average. This
distinction is now better explained in lines 46-48. We have now also
commented on the interpretation of Cho11 in the new paragraph on page 9
around Eq.(16), starting with "It should be noted ...", where we write at
the end of the paragraph "Earlier interpretations in terms of Whistler
waves (Cho11) seem therefore artificial and obscured the relevant
interpretation of the magnetic field as a quantity with dimensions of
$\m^2\s^{-1}$; see Eq. (2.6)."

> A final question concerns additional effect, such as rotation, which is
> important for the geo-dynamo problem. In principal, is it possible to
> generalize this invariance to rotating MHD?

Rotation generally suppresses turbulence and thereby the nonlinearity
essential for any inverse cascade to occur. This is now described in
the new subsection 2.4 on page 7, along with a new Figure 4.